# Epithelial-Mesenchymal Transition and Senescence in the Retinal Pigment Epithelium of *NFE2L2/PGC-1α* Double Knock-Out Mice

**DOI:** 10.3390/ijms22041684

**Published:** 2021-02-08

**Authors:** Janusz Blasiak, Ali Koskela, Elzbieta Pawlowska, Mikko Liukkonen, Johanna Ruuth, Elisa Toropainen, Juha M. T. Hyttinen, Johanna Viiri, John E. Eriksson, Heping Xu, Mei Chen, Szabolcs Felszeghy, Kai Kaarniranta

**Affiliations:** 1Department of Molecular Genetics, Faculty of Biology and Environmental Protection, University of Lodz, Pomorska 141/143, 90-236 Lodz, Poland; janusz.blasiak@biol.uni.lodz.pl; 2Department of Ophthalmology, University of Eastern Finland, 70210 Kuopio, Finland; ali.koskela@uef.fi (A.K.); mikko.liukkonen@uef.fi (M.L.); johanna.ruuth@uef.fi (J.R.); elisa.toropainen@uef.fi (E.T.); Juha.Hyttinen@uef.fi (J.M.T.H.); johanna.viiri@uef.fi (J.V.); 3Department of Orthodontics, Medical University of Lodz, 92-217 Lodz, Poland; elzbieta.pawlowska@umed.lodz.pl; 4Turku Bioscience Centre, University of Turku and Åbo Akademi University, 20520 Turku, Finland; john.eriksson@utu.fi; 5Cell Biology, Faculty of Science and Technology, Åbo Akademi University, 20500 Turku, Finland; 6The Wellcome-Wolfson Institute of Experimental Medicine Queen’s University Belfast, 97 Lisburn Rd, Belfast BT9 7BL, UK; heping.xu@qub.ac.uk (H.X.); mei.chen@qub.ac.uk (M.C.); 7Institute of Biomedicine, University of Eastern Finland, Yliopistonranta 1 A, 70211 Kuopio, Finland; szabolcs.felszeghy@uef.fi; 8Institute of Dentistry, University of Eastern Finland, Yliopistonranta 1 C, 70211 Kuopio, Finland; 9Department of Ophthalmology, Kuopio University Hospital, 70210 Kuopio, Finland

**Keywords:** age-related macular degeneration, AMD, epithelial-mesenchymal transition, senescence, *NFE2L2/PGC-1α* dKO mice

## Abstract

Age-related macular degeneration (AMD) is the most prevalent form of irreversible blindness worldwide in the elderly population. In our previous studies, we found that deficiencies in the nuclear factor, erythroid 2 like 2 (*NFE2L2*) and peroxisome proliferator-activated receptor gamma coactivator 1-α (*PGC-1α*) genes caused AMD-like pathological phenotypes in mice. In the present work, we show hijacked epithelial-mesenchymal transition (EMT) due to the common loss of *PGC-1α* and *NFE2L2* (double knock-out, dKO) genes in aged animals. The implanted area was assessed by histology, immunohistochemistry and transmission electron microscopy. Confocal microscopy revealed altered regions in the filamentous actin ring. This contrasted with hexagonal RPE morphology in wild-type mice. The ultrastructural RPE features here illustrated loss of apical microvilli, alteration of cell-cell contact, loss of basal in-folding with deposits on Bruch’s membrane, and excessive lipofuscin deposition in dKO samples. We also found the expression of epithelial-mesenchymal transition transcription factors, such as Snail, Slug, collagen 1, vimentin and OB-cadherin, to be significantly different in dKO RPEs. An increased immunoreactivity of senescence markers p16, DEC1 and HMGB1 was also noted. These findings suggest that EMT and senescence pathways may intersect in the retinas of dKO mice. Both processes can be activated by damage to the RPE, which may be caused by increased oxidative stress resulting from the absence of *NFE2L2* and *PGC-1α* genes, important for antioxidant defense. This dKO model may provide useful tools for studying AMD pathogenesis and evaluating novel therapies for this disease.

## 1. Introduction

Epithelial–mesenchymal transition (EMT) is a process in which polarized epithelial cells, normally interacting with the basement membrane through their basal surface, acquire a mesenchymal-like phenotype [1]. Cells that undergo EMT are characterized by increased migratory potential, resistance to apoptosis and the production of components of the extracellular matrix (ECM). The completion of EMT is associated with the degradation of the basement membrane and the migration of newly transformed mesenchymal cells from the epithelial layer (reviewed in [2]).

EMT is needed in normal embryogenesis (type 1 EMT) and wound healing (type 2), but it also occurs in some pathological conditions, including cancer, in which EMT plays a crucial role in metastasis (type 3) (reviewed in [3,4]). EMT is controlled by directed interactions among many proteins in several signal transduction pathways with core EMT transcription factors, such as Snail, Slug, ZEB1, ZEB2 and Twist [5]. All these factors regulate the expression of the cadherin family of proteins that modulate cell adhesion [6]. The Snail superfamily of zinc-finger transcription factors regulates EMT as its members SNAI1 (Snail), SNAI2 (Slug) and SNAI3 (Smuc) are involved in the transcriptional inhibition of E- and VE-cadherins [7]. Vimentin is expressed in mesenchymal cells and is frequently used as a molecular marker of EMT reprogramming [8].

Type 2 EMT occurs in organ fibrosis and is mediated by inflammatory cells and fibroblasts that release different inflammatory mediators and components of the extracellular matrix (ECM), including collagens, elastin, laminins and tenascins [9]. In the eye, subretinal fibrosis is a marker of advanced stage of choroidal neovascularization (CNV) in wet age-related macular degeneration (AMD) which ultimately results in vision loss (reviewed in [10]). In AMD, retinal pigment epithelium (RPE) cells degenerate, losing their mutual contact and cell polarity, causing them to migrate into the retinal and subretinal RPE space [11]. However, most of these degenerated cells do not die, but rather transform into mesenchymal cells to survive in the severe environment associated with AMD progression. Such a reversible transformation can be achieved through EMT [3]. The process of subretinal fibrosis in the eye displays several features of scar formation seen in other organs. Profibrotic cells migrate to and proliferate at the Bruch’s membrane-RPE complex, where they form subretinal fibrosis tissue with the involvement of other cells, mainly matrix-producing mesenchymal cells [12]. EMT in mouse RPE was induced by the injection of activated macrophages into the subretinal space [13].

Progression of AMD to its advanced stage, resulting from oxidative stress, RPE degeneration and low-level chronic inflammation, displays some common features with aberrant wound healing [14]. Type 2 EMT, typical for wound healing, can be activated by impairment in lysosomal functions, including autophagy [15]. Impaired autophagy contributes to the physiological and pathological processes of RPE, including stimulation of type 3 EMT [16]. Furthermore, dysregulated autophagy can contribute to AMD pathogenesis [4,17,18]. As autophagy itself is a complex process, its involvement in AMD pathogenesis is not completely clear. Mitter et al. suggested that oxidative stress-induced autophagy in AMD may be dysregulated in two ways, depending on whether the stress is acute (autophagy increase) or chronic (autophagy decrease) [19]. AMD is associated with the accumulation of lipofuscin in the RPE. Lipofuscin contains lysosomal insoluble pigment granules that remain after lysosomal digestion [17]. Rapamycin, an autophagy inducer, was reported to decrease lipofuscin accumulation in RPE cells [19].

Although the general role of EMT in AMD, especially its wet form, seems to be well-established, the mechanism of EMT in dry AMD is less known. It was shown that βA3/A1-crystallin, encoded by the *Cryba1* gene, played an important role in the EMT in dry AMD in human and murine RPE cells [20]. The βA3/A1-crystallin protein, belonging to the β/γ-crystallin family, localizes to the lysosomal lumen of RPE cells and influences the lysosomal function of RPE cells, including phagocytosis and autophagy [21]. Loss of βA3/A1-crystallin resulted in the upregulation of Snail and vimentin, downregulation of E-cadherin and increased migration of RPE cells [20]. These results suggest that EMT may be associated with autophagy impairment in dry AMD. Decreased autophagy is associated with cell phenotype shifting to senescent cells that contribute to a loss of tissue homeostasis [22,23]. 

Current AMD therapy faces many challenges. The most prevalent form of AMD, dry AMD (85% of cases), is currently untreatable. Only wet AMD can be treated with intravitreal anti-vascular epithelial growth factor (VEGF) inhibitors that slow disease progression. The life-long monthly injections are a burden to patients and health care systems. Moreover, intravitreal injections are invasive procedures with some potential side effects, such as hemorrhages and inflammatory reactions. Total worldwide costs for AMD treatment are estimated to be 350 billion USD per year (AMD Alliance International. http://www.brightfocus.org/sources-macular-degeneration-facts-figures (accessed on 20 January 2021)).

Some successes in gene therapy and stem cell-based treatment have not been implemented in clinical practice. Retinal drug therapy is hampered by ineffective and/or short-acting drug delivery to the targets. Eye drop instillation does not provide sufficient drug bioavailability and the blood-retinal barrier limits the penetration of systemic drugs. For most drugs, intravitreal injections are not useful, because small molecules are eliminated from the eye within two–three days. Many protein drugs have a narrow therapeutic index and they cannot reach their biological targets [24]. Therefore, the treatment of AMD is very limited and, in many cases, it can be considered as an incurable disease. One of the reasons is the complexity of AMD pathogenesis and our incomplete knowledge of its underlying mechanisms. Therefore, studies on molecular aspects of AMD pathogenesis are needed as their results can contribute to the identification of new therapeutic targets.

We have genetically modified mice that bear knockouts in the nuclear factor, erythroid 2 like 2 (*NFE2L2*) and peroxisome proliferator-activated receptor gamma coactivator 1-α (*PGC-1α*) genes (*NFE2L2/PGC-1α* double knock-out/dKO mice) [25]. These mice show many features of dry AMD, including age-dependent RPE degeneration, increased oxidative stress, damaged mitochondria, changes in protein ubiquitination and autophagy, photoreceptor dysmorphology and vision loss. These changes suggest that the dKO mice serve as a model organism of dry AMD. In the present work we investigated the ultrastructural changes in the retinas of *NFE2L2/PGC-1α* dKO and wild-type (WT) mice. We also investigated the immunoreactivity of mesenchymal markers Snail (SNAI1), Slug (SNAI2), collagen 1, vimentin and OB-cadherin (Cadherin-11) and the senescence markers p16 (an Ink4 family cyclin dependent kinase/tumor suppressor), DEC1 (deleted in esophageal cancer 1) and HMGB1 (high mobility group box 1) in the RPE cells of the dKO and WT mice. 

## 2. Results

### 2.1. Inactivation of NFE2L2/PGC-1α Affects the Retinal Pigment Epithelium Nuclei Density

Although we recently published that the *NFE2L2/PGC-1α* knock-out mouse could be an animal model of dry AMD, we did not carry out detailed quantitative analysis of cellular morphological parameters [25]. Therefore, in our current work, we first set out to quantitatively elucidate how these pivotal genes affect the retinal pigment epithelium morphology at light microscopic level.

In the toluidine stained transverse semithin sections (Figure 1A–D), more RPE nuclei per field of view were found in the one-year-old WT sample compared to the dKO mice. This difference was uniform throughout the analyzed RPE layer. When measurements of the linear packing density of nuclei were made as an indirect estimate of RPE cell number, there was a definitive decrease in the number of RPE cells per unit length in the *NFE2L2/PGC-1α* dKO RPE compared to the WT sample.

### 2.2. Loss of the NFE2L2/PGC-1α Genes Affects RPE Monolayer Arrangment and Cell–Cell Junctional Complexes

Next, to test whether the change in the nuclear linear packing density in the dKO mutant RPE semithin sections reflects altered cellular density and/or arrangement in the retinal epithelial layers, Alexa Fluor 568-phalloidin stained retinal flat-mount samples were analyzed.

Figure 2A,C compare images obtained from preparations of one-year-old wild-type and age-matched dKO mutant littermate flat-mount retina samples. The RPE monolayer from WT animals is comprised of a relatively normal cobblestone array of cells that are hexagonal in shape and in overall arrangement. In addition, clear differences in cellular density were apparent in dKO mutant retinal pigment epithelium, which appears less crowded. We also aimed to examine the expression of tight junction cytoskeletal protein (F-actin) with fluorescein phalloidin antibody in the flat-mount RPE samples of WT and *NFE2L2/PGC-1α* dKO mice. We observed several altered phalloidin staining regions of filamentous actin ring structures in dKO cells compared to their WT counterparts (Figure 2A,C). The magnified high-power images show irregular intercellular apical junctions in dKO RPE cells (Figure 2C, yellow arrowheads) compared to WT RPE (Figure 2A).

As alteration of cell–cell adhesion might promote cellular phenotype changes, we applied transmission electron microscopy (TEM) to examine the cell–cell contacts in the RPE layer to further understand the possible role of the loss of *NFE2L2* and *PGC-1α* genes in EMT. Therefore, first, TEM was used to compare the cellular junctions of the RPE in WT and *NFE2L2/PGC-1α* dKO mice. WT animals showed normal tight junctions (Figure 2B) in contrast to the dKO samples, in which irregularly shaped apical tight junctions of the RPE were observed, suggesting the breakdown of the outer barrier (Figure 2D). RPE apicolateral zone and photoreceptor outer segments from dKO samples accumulated more melanin pigment granules at the apical RPE cells’ cytoplasm (Figure 2D, inset) compared to WT animals (Figure 2B, inset).

### 2.3. Ultrastructural Alteration at Basal Compartment of Aged NFE2L2/PGC-1α Mice

Next, to answer the question as to whether the loss of *NFE2L2* and *PGC-1α* genes induces EMT-like changes in RPE cells, we evaluated the ultrastructural alterations in cellular architecture by transmission electron microscopy.

As shown in Figure 3, in WT and young (three-months-old) dKO samples the RPE structures were well-developed with retained cellular polarity in contrast to the one-year-old dKO samples where, after *NFE2L2/PGC-1α* gene knockdown, many RPEs generally lost their basal infoldings and showed elongated basal laminar deposits (Figure 3D, arrow).

### 2.4. Loss of Apical Microvilli of RPEs in Aged NFE2L2/PGC-1α Mice

Transmission electron microscopy was further used to examine the effect of the loss of *NFE2L2/PGC-1α* on the apical compartments of RPEs. In particular, we analyzed the ultrastructure of RPE photoreceptor interface.

As shown in Figure 4A, WT RPEs have densely arranged microvilli (arrowheads). They are closely associated and have tight contact with photoreceptor outer segments. This organization is typical of cells with a well-developed apical polarity. This organization of cellular extension was strongly affected in dKO RPE samples (Figure 4B, arrow).

Using morphometric analysis, we were able to assess the RPE height in one-year-old dKO mice. In close vicinity to the optic nerve entrance in the innermost layer of the retina, the height of the centrally located RPE was markedly decreased (Figure 4C). Thus, knockout of *NFE2L2* and *PGC-1α* genes in mouse RPE cells in vivo triggered morphological, mesenchymal-like ultrastructural changes.

### 2.5. NFE2L2/PGC-1α dKO Animals Show Increased Immunoreactivity of the Mesenchymal Markers Snail, Slug, Vimentin and OB-Cadherin, but a Decreased Signal of Collagen

An increased immunoreactivity of the mesenchymal markers Snail, Slug, vimentin and OB-cadherin in the retina of dKO mice as compared to the WT animals was observed (Figure 5).

A low signal was found to be localized at the cytoplasm in the RPE of the retina in WT samples. The re-localization of these markers from the cell surface to the cytoplasm, especially around the nucleus of dKO animals, was a characteristic sign for EMT. Similar effects were detected for the remaining mesenchymal markers, Snail and Slug. In contrast to the EMT markers, a decreased immunoreactivity of collagen was observed.

### 2.6. NFE2L2/PGC-1α dKO Mice Show Increased Immunoreactivity of the Senescence Markers p16, DEC1 and HMGB1

As we aimed to study the hallmarks of retinal senescence, we first compared the accumulation of lipofuscin (LF) granules in the cytoplasm of RPE cells obtained from one-year-old WT and dKO mice. To monitor whether the loss of *NFE2L2* and *PGC-1α* genes affects the deposition of this nondegradable fluorescent material, we used confocal microscopy to analyze the deposition of LF in the RPE of one-year-old WT and dKO mice (Figure 6). We detected more LF-like autofluorescence (488-nm excitation) in the RPE of one-year-old dKO RPE compared with their WT littermates (Figure 6A,B). Next, we analyzed the number of LF and the area fraction of LF granules per 100 μm^2^ from electron micrograph sections of RPE. For the dKO RPEs. The fraction was approximately 65% higher than that seen in the WT littermates. These observations indicated that lipofuscin granules were formed and extensively accumulated within dKO RPE, because of metabolic changes occurring due to the loss of *NFE2L2* and *PGC-1α* genes.

Finally, we studied EMT’s connection to some senescence markers (Figure 7).

To further investigate whether the common knockdown of *NFE2L2/PGC-1α* genes induce cellular senescence in RPE layer we used three different senescence markers to detect it. Interestingly, it was detected with semiquantitative data analysis that in one-year-old dKO samples showed upregulation of p16, DEC1, HMGB1 markers in RPE cells compared with the WT littermates, suggesting that these markers are associated with *NFE2L2/PGC-1α* -induced cellular senescence in RPEs.

## 3. Discussion

Although several studies have revealed different molecular mechanisms that drive RPE dysfunction and lead to epithelial-mesenchymal transition, the effect of the common loss of *NFE2L2/PGC-1α* remains to be determined in vivo. In the current work, we set out to elucidate how these pivotal genes affect RPE cellular polarity and tight junctions. We have also discovered experimental evidence that aberrant mesenchymal marker (Snail, Slug, vimentin and OB-cadherin) expression was critically associated to the accumulation of increased senescence marker (p16, DEC1 and HMGB1) expression, which together might drive EMT in aged RPE cells in dKO animals. Collectively, our results suggest that the loss of *NFE2L2* and *PGC-1α* might influence EMT of RPEs. However, the precise molecular mechanisms remain elusive.

We recently suggested that the *NFE2L2/PGC-1α* double knockdown mouse could be a potential animal model of dry AMD and reported that the RPE cells from these one-year-old dKO animals appear smaller than those of their normal littermates [25]. Now, using more detailed morphometric analysis, we have been able to confirm these observations quantitatively. Our recent quantitative data indicates that the one-year-old dKO RPE is smaller in its apical-to-basal dimension than that of the RPE from age-matched WT littermates. Notably, the linear nuclear density and the monolayers from these mutant animals exhibited a lower nuclear and somewhat decreased cellular density, respectively. These may suggest that, concomitantly with an enhancement in cellular death also reported earlier [25], the average lateral dimension of some of the individual dKO RPEs increases, notwithstanding the relative decrease in its thickness. Such morphological changes in the average RPE cell volume might be indicative of a mechanism whereby dKO RPE cells are able to balance their growing/flattening rate in order to cover a defined tissue size and to try to maintain and support photoreceptor function. Nonetheless, parallel with these morphological changes discussed above, in dKO mice tight junctions were found to be disorganized in RPE flat-mount samples and a loss of apical microvilli and basal infoldings of the RPE were also observed, which can also be partial hallmarks of epithelial phenotype changing to mesenchymal: maybe these polarity and cell–cell contact changes are the key processes for quiescent RPE cells to re-enter the cell cycle.

Lipofuscin is a complex aggregate of pigmented material resulting from incomplete lysosomal degradation of phagocytosed photoreceptor outer segments. The precise composition of lipofuscin is not clear; however, it is also found in senescent cells. Detection of lipofuscin is one of the feasible methods used for identification of senescent cells both in vitro and in vivo [23]. It is worth mentioning, that the present study also documents that *NRF-2/PGC-1α* genes double knockdown induced lipofuscinosis. This was evidenced by autofluorescence and it was confirmed by ultrastructural examination. Our data demonstrate that lipofuscin accumulation positively correlates with common loss of *NFE2L2/PGC-1α* genes, and that the lipofuscin signal overlaps with other established senescent markers DEC1, HGBM1 and Snail in the RPE of the *NFE2L2/PGC-1α* dKO samples.

A normal epithelium does not usually display features of EMT, but its damage may induce EMT [6]. The substantial changes observed in the *NFE2L2/PGC-1α* dKO RPE could be recognized as damage caused to the retina by EMT. Our results confirm recent findings of the Saint-Geniez laboratory, which reported that the loss of *PGC-1α* in mouse RPE cells resulted in retinal degeneration and EMT [26]. However, they inactivated *PGC-1α* specifically in the RPE. AMD is not strictly localized to the choriocapillaris, RPE or the retina, since many systemic regulators affect its development [27]. As we and others have shown, senescence could be an important factor of AMD pathogenesis [28,29,30]. Moreover, although established mainly in cancer, it is generally accepted that senescence intersects with EMT [31,32]. We previously showed, and have now confirmed in the present research, that *NFE2L2/PGC-1α* dKO mice display several features of damaged RPE. 

Although apparently clear, the link between aging and senescence is not direct, as it consists of many possible pathways [25]. Aging and aging-related diseases are reported to be underlined by the accumulation of senescent cells, and their selective elimination was reported to extend the lifespan of mice [33,34]. Senescent cells, which are characterized by senescence-associated secretory phenotype (SASP), release factors that can initiate or increase existing low-grade inflammatory processes, which are of special relevance to aging and age-associated diseases both in general and in AMD in particular [35,36]. Senescence-accelerated OXYS rats are an established model of premature aging and of some neurodegenerative diseases, including AMD [37,38].

Our work has several limitations. Although key mesenchymal and senescence markers were analyzed, larger proteomic analysis is an option to identify common proteins of EMT and senescence in RPE of the *NFE2L2/PGC-1α* dKO mice. Secondly, a question which might arise from our study and needs to be further clarified is whether lipofuscin accumulation is a mere consequence of the induction of the senescence program or whether the increased number of lipofuscin aggregates contributes causally to the development of senescence in RPE cells of the *NFE2L2/PGC-1α* dKO mice. However, several indirect and direct studies may extend our observations, including epigenetic profiling of RPE cells and RPE de-differentiation [26,38].

In conclusion, the *NFE2L2/PGC-1α* dKO mice present gross morphological changes in the outer retina and RPE that may be underlined by the increased oxidative stress sustained by the mice due to their impaired antioxidant defense caused by the deletion of two genes important for such functions. These changes may be identified by retinal damage, and EMT and senescence pathways may be activated by the cells to prevent further damage to the retina (Figure 8). With this study, we have extended the range of AMD-related effects that can be studied with the *NFE2L2/PGC-1α* dKO mice and, thus, targeting senescence might be a promising treatment for AMD.

## 4. Materials and Methods

### 4.1. Animals

Transgenic C57BL/6J mice with global knockouts in the *NFE2L2* and/or *PGC-1α* genes were generated and checked for their genotypes as described previously [25]. All animal experiments were performed according to the protocols that were in agreement with the ARVO Statement for the Use of Animals in Ophthalmic and Vision Research and approved by the Project Authorization Board of Regional Administrative Agency for Southern Finland (ESAVI/8893/04.10.07/2014). The animals were maintained in a 12/12 h light-dark cycle at constant temperature (22 ± 1 °C) and had free access to drinking water and standard pellet chow. Mice (6) weighing between 18–24 g were used in this study.

### 4.2. Flat Mount Phalloidin Immunostaining

To confirm identification of F-actin by phalloidin staining, retinal flat mount samples were prepared, as previously described [25]. Briefly, after deep anesthesia, mice were sacrificed, and enucleated eyes were fixed in 2% paraformaldehyde for 2 h. The RPE complex was incubated in permeabilization solution (1% Triton-X 100 and 0.3% BSA in PBS) at room temperature (RT) for 2 h. Then, the eyes were incubated at RT for 2 h with primary antibody Alexa Fluor 568 Phalloidin (1:100, Thermo-Fisher Scientific, Waltham, MA, USA). After washing with PBS, the RPE/choroid complex was mounted with Fluoromount™ Aqueous Mounting Medium (Sigma Aldrich, St. Louis, MO, USA) and observed under confocal microscope (Eclipse TE200-U; Nikon UK Ltd., 1 The Crescent, Surbiton, UK). Z-stack confocal images of RPE flat mounts were reconstructed using the NIS Element (Nikon, Shinagawa, Tokyo, Japan) software. The border of each RPE cell was outlined based on Phalloidin (F-actin) staining.

### 4.3. Eyeballs for Immunohistochemical Analysis

The mice were sacrificed, and eyes were collected immediately and fixed as above. From paraffin embedded blocks, 3 μm-thick parasagittal serial sections were cut with a microtome (Leica, Heidelberg, Germany). After the random selection of slides from each individual specimen at the level of optic entrance, sections were dewaxed and rehydrated and the labeling of the retina was performed according to previously published methods, with minor modifications [38]. The antibodies used are shown in Table 1.

Sections for Snail, Slug and Collagen 1 immunohistochemistry were pre-treated with 1× citrate buffer for 5 min in 90 °C; otherwise, all the sections were processed according to the same protocol. Sections were quenched for 10 min with 0.1 M glycine-PBS and permeabilizated twice for 10 min with 0.1% Triton-TBS. After rinsing, sections were blocked with 20% goat serum in TBS for 30 min. Primary antibodies were incubated overnight at 4 °C. After primary antibody incubation, the sections were first rinsed and then incubated in dark with secondary antibody Alexa Fluor 594 diluted 1:500 in TBS for 3 h. After washing the sections three times in dark, fluorescent nuclear marker DAPI (4,6-diamidino-2-fenilindole, dihydrochloride) (Sigma Aldrich, St. Louis, MO, USA) was diluted 1:10,000 in TBS and incubated for 30 min in RT. After rinsing, slices were covered with Mowiol mounting media (Sigma Aldrich, St. Louis, MO, USA).

### 4.4. Confocal Scanning and Imaging

Eye sections were examined with a confocal microscope (Zeiss AX10 Imager A2, Zeiss, Göttingen, Germany). Images were recorded sequentially from the green, red and far-red channels on optical slices of 0.1 μm of focal thickness using a 63× oil immersion objective (NA:1.42, Plan Apochromat). The microscopy settings were identical for all scans and kept constant during imaging. Representative high-power micrographs were taken with ZEN black software and processed with Adobe Photoshop. In all imaging procedures, gamma adjustment was performed on the whole image in order to maintain appropriate contrast. Representative images from mice (*n* = 6) were used for the demonstration of each observation.

### 4.5. EMT Biomarkers

The immunohistochemical results were examined by three independent observers searching for immunoreactive RPE cells. The different marker distribution patterns were analyzed using ImageJ software as described below. For quantitation of the different EMT related proteins, RPE cells per animal of interest were manually designated as regions of interest (ROI). Special care was taken to select ten representative areas per section from each individual sample in the vicinity of the optic nerve. To analyze photomerged images, ImageJ software was used to quantify the level of the positive immunoreaction for each marker studied (intensity of B&W pseudo-color) within each ROI. This was done by determining the pixel density of each signal and then converting each individual pixel into a numerical value between 0 (no immunohistochemical reaction or signal at background noise level, which was determined to be below 20) and 255 (highest staining intensity).

### 4.6. Transmission Electron Microscopy Studies

For transmission electron microscopy (TEM), eight eyes from four mice were used from 12-month-old WT and littermate dKO samples. Samples were processed and analyzed according to our previous publication [25]. Briefly, the specimens were washed in 0.1 M cacodylate buffer containing 3.7% (*w*/*vol*) saccharose and sequentially dehydrated in 30% (*vol*/*vol*) ethanol at 4 °C, 50% (*vol*/*vol*) ethanol at 0 °C, and 70% (*vol*/*vol*) ethanol at −20 °C. The specimens were then immersed in a 1:1 mixture of 70% ethanol and medium grade LX112 resin (Ladd Research Industries, Williston, VT, USA) at 4 °C and several changes of pure resin at 4 °C and RT, and finally polymerized in gelatin capsules for 1 day at 45 °C. One-micron semi-thin sections were cut with a Reichert Ultracut-E microtome (Leica Microsystems Inc., Buffalo Grove, IL, USA), stained with 1% toluidine blue and examined with a light microscope to find the localization of interest (i.e., RPE) prior to further TEM sectioning. Ultrathin sections were cut with a microtome (Ultracut; Leica, Bensheim, Germany) and mounted on uncoated nickel grids. Subcellular structures of RPE were assessed using JEM-1010 TEM (Tokyo, Japan) transmission electron microscope. Similarly, as conducted with the histological samples, representative areas per sample were selected by collecting RPEs from each individual one-year old WT and dKO samples which were close to the optic nerve. Representative images from mice (*n* = 6) were used for the demonstration of each observation.

### 4.7. Detection and Quantitative Analysis of the Number and Area of Fractional Lipofuscin in RPE

Detection of lipofuscin was performed as previously described [25]. The samples were examined in a laser scanning confocal microscope (Zeiss AX10 Imager A2, Zeiss, Göttingen, Germany) with sequential scanning and detection of dot-like lipofuscin granules, respectively, followed by merging and saving of the images. For quantitative analysis, from each TEM image, using the open source ImageJ software (http//:imagej.nih.gov (accessed on 20 January 2021); NIH, USA) the amount of lipofuscin in 100 μm^2^ and fractional lipofuscin granules were measured by obtaining the area occupied by lipofuscin over the area in 100 μm^2^ occupied by cytoplasm. One eye from each animal was included, and each eye was averaged (*n* = 6 animals).

### 4.8. RPE Cell Density and Morphology Analysis

Samples from one-year-old WT and *NFE2L2/PGC-1α* dKO mice were compared for changes in RPE cell density. For analysis, semithin tissue sections were prepared as described in the Section 4.5. The semithin toluidine blue sections were obtained in a vertical plane through the middle of the eye. One-micron semithin sections were examined with light microscope. Analyses were made along the length of the epithelium within a single section, excluding regions within 300 μm of the optic nerve and areas exhibiting outer nuclear layer dysplasia. As an estimate of cellular density, counts of nuclei contained within 100 μm expanses of the retinal pigment epithelium layer were determined at the same sites and location to preclude the effect of cellular density variability from central to peripheral regions. Samples from one-year-old WT and *NFE2L2/PGC-1α* dKO mice were compared for changes in RPE cell size. ImageJ 1.43 freeware (http//:imagej.nih.gov (accessed on 20 January 2021); National Institutes of Health, Bethesda, MD, USA) was used for RPE cell size analysis. For a given RPE, tens of cells were assessed and averaged to generate a single result for central RPE. Final data is presented per each group and were assessed for statistical comparison.

## 5. Data Analysis

Statistical analyses were performed with IBM SPSS Statistics for Windows (version 24.0, Armonk, NY, USA). Shapiro-Wilk test of normality was conducted and due to data not being in a normal distribution (*p* < 0.05), and non-parametric Mann-Whitney U-test was used for comparisons between groups.

## Figures and Tables

**Figure 1 ijms-22-01684-f001:**
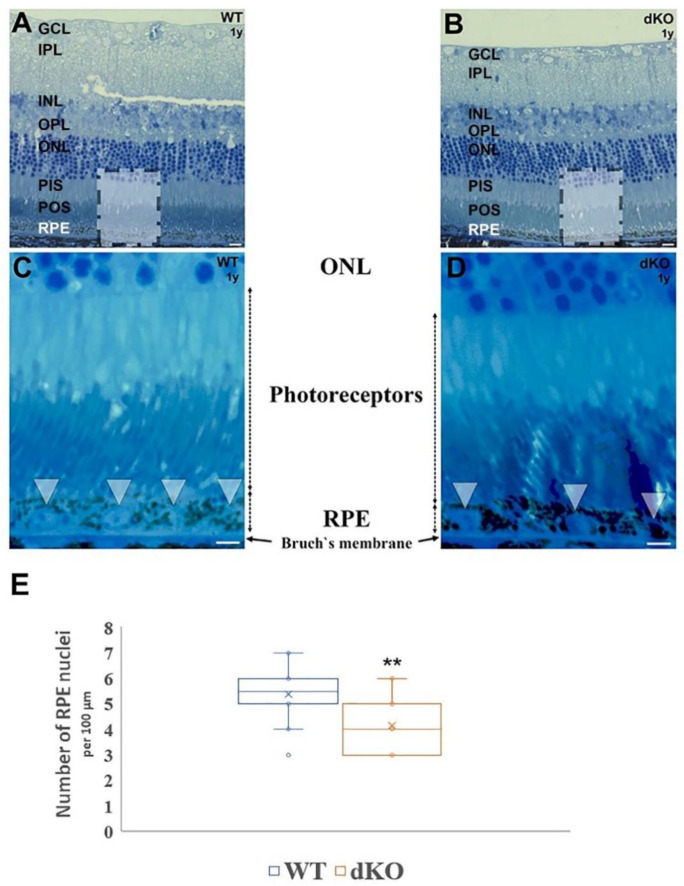
Light micrographs of toluidine stained semithin retinal sections derived from one-year-old wild-type (WT) and *NFE2L2/PGC-1α* dKO mice. Retinal cross sections were prepared from one-year-old WT control mice (**A**,**C**) and dKO samples (**B**,**D**) and plastic sections were stained with toluidine blue. Boxed areas are shown with high power in (**C**,**D**). The RPE cell layer from WT animals is thicker, with a higher number of nuclei (arrowheads in (**C**)) compared to age-matched dKO littermates (arrowheads in (**D**)). In the *NFE2L2/PGC-1α* dKO mice RPE layer is flattened. Melanin granules tend to occupy more of their cytoplasm. (**E**) illustrates the decrease in the density of RPE nuclei for double knockout animals, relative to the age-matched WT littermates (*n* = 30). ** *p* < 0.001 (Mann-Whitney U-test). Abbreviations: GCL: ganglion cell layer; IPL: inner plexiform layer; INL: inner nuclear layer; OPL: outer plexiform layer; ONL: outer nuclear layer; PIS: photoreceptor inner segment; POS: photoreceptor outer segment; RPE: retinal pigment epithelium. Scale bars: 5 µm.

**Figure 2 ijms-22-01684-f002:**
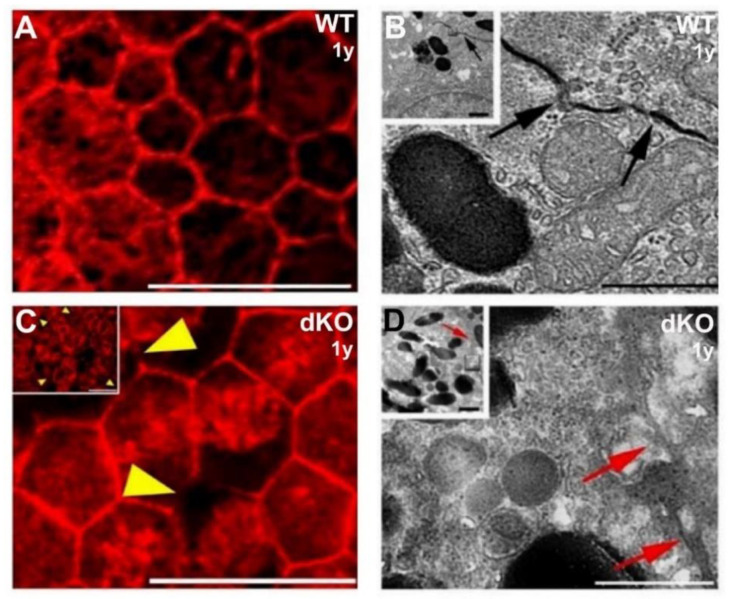
Structural changes of the apicolateral retinal pigment epithelium (RPE) cell–cell junctions in 1-year-old *NFE2L2/PGC-1α* dKO mice as compared with their wild-type (WT) counterparts. The pictures shown in (**A**,**C**) are from phalloidin-based immunofluorescent flat mount confocal microscopy, while transmission electron microscopy (TEM) images are presented in (**B**,**D**). Phalloidin was used to visualize filamentous actin rings on flat-mount samples and indirectly examine the cellular density and arrangement. The WT retinal epithelium consists of typical hexagonal cells (**A**), while dKO RPE cells are, by contrast, less regular in their outline. While cellular density can vary somewhat, regions of the WT RPE display higher cellular density. In 1-μm-thick confocal sections of phalloidin-labeled tissues, apical cell boundaries are marked by rings of actin filaments associated with intercellular junctions. Yellow arrowheads indicate multiple locations of the altered phalloidin staining for the apical aspect of the RPE complex. Inserts in (**B**,**D**) illustrate the RPE apicolateral zone and the high power TEM images show the profile of lateral RPE cell–cell contact (black (**B**) and red (**D**) arrows). Scale bars: 20 µm (**A**,**C**) and 1 µm (**B**,**D**).

**Figure 3 ijms-22-01684-f003:**
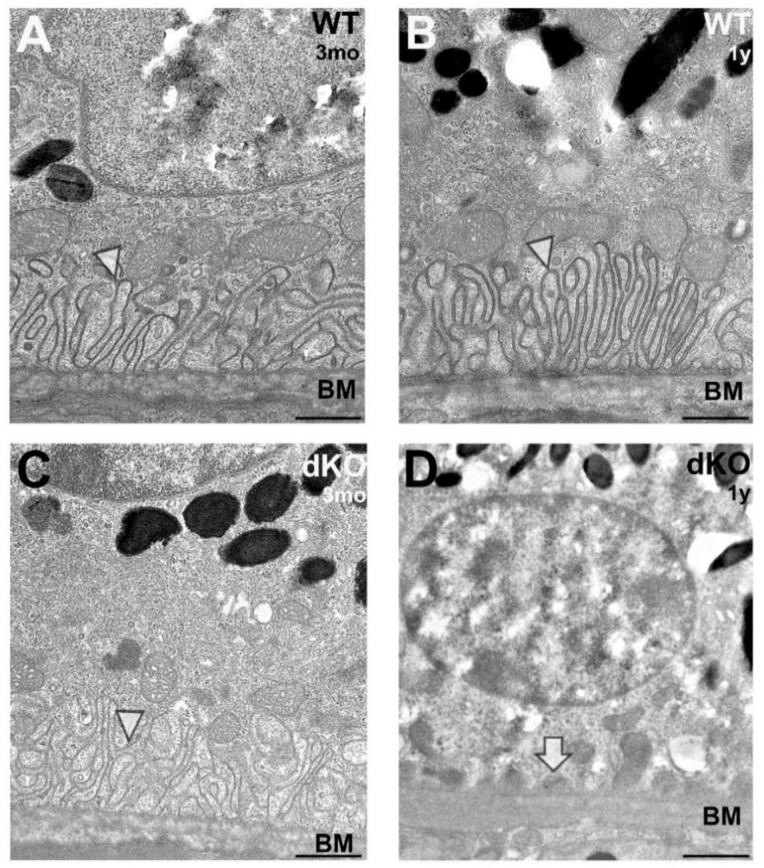
Structural changes of the basal compartment of the retinal pigment epithelium (RPE) in *NFE2L2/PGC-1α* double knockout (dKO) mice as compared to their age-matched wild-type (WT) counterparts. Representative high power transmission electron microscopy (TEM) images of the samples of three-months- (**A**,**C**) and one-year-old (**B**,**D**) WT and dKO animals are shown. RPE of WT mice at both time points (**A**,**B**) and the three-months-old dKO samples (**C**) show normal basal arrangement: well-organized infoldings above Bruch’s membrane, supporting intracellular organelles (infoldings indicated by arrowheads) and typical thickness of Bruch’s membrane (BM). In contrast, many of the one-year-old *NFE2L2/PGC-1α* dKO mice RPEs show ultrastructural signs of loss of the basal infoldings with basal linear deposits (**D**; arrow), resulting in unsupported intracellular organelles above Bruch’s membrane. Scale bars: 1 µm.

**Figure 4 ijms-22-01684-f004:**
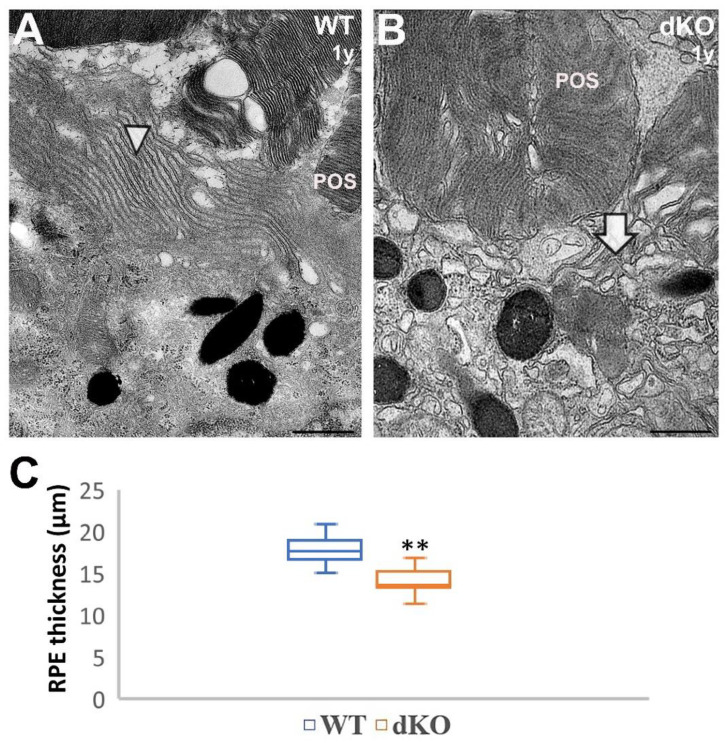
Loss of apical microvilli and flattened RPE in one-year-old *NFE2L2/PGC-1α* dKO mice as compared with their age-matched wild-type (WT) littermates. Panel 4 **A** shows the apical portion of WT RPE, where microvilli (arrowhead) are arranged normally, surrounding the photoreceptor outer segments (POS). In contrast, many of the apical portion of RPEs from dKO mice showed loss of apical microvilli: only a few microvilli on the apical surface were evident (**B**; arrow), with an increased density of melanin granules. Total RPE height from TEM graphs was measured in the region located 300 µm from the optic nerve head (**C**). Combined results from *n* = 6 animals, ** *p* < 0.001 (Mann-Whitney U-test). Scale bars: 1 µm (**A**,**B**).

**Figure 5 ijms-22-01684-f005:**
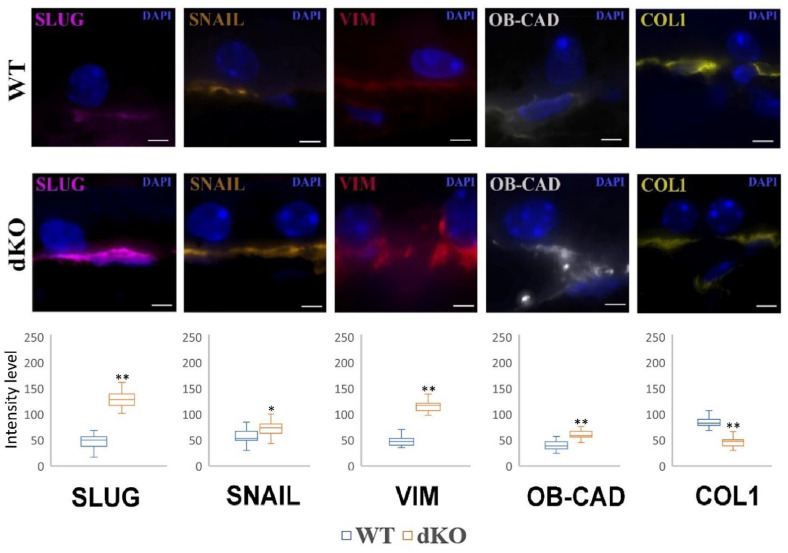
Confocal microscopy analysis of the immunoreactivity of mesenchymal markers in the retinal pigment epithelium (RPE) of one-year-old *NFE2L2/PGC-1α* dKO mice as compared with their wild-type (WT) counterparts. Semiquantitative comparative densitometric analysis done by ImageJ of the immunoreactivity of Slug, Snail, vimentin, OB-cadherin (OB-CAD) and collagen-1 (COL1) in 1-μm parasagittal sections of RPE. Alexa Fluor secondary antibodies were used to detect specific mesenchymal markers. DAPI was used to stain the nuclei of RPE cells. Scale bar: 5 μm. The number of puncta specific to a particular marker is presented in plots in the lower panels. Twenty-five pictures were analyzed for each animal. * *p* < 0.05, ** *p* < 0.001 (Mann-Whitney U-test).

**Figure 6 ijms-22-01684-f006:**
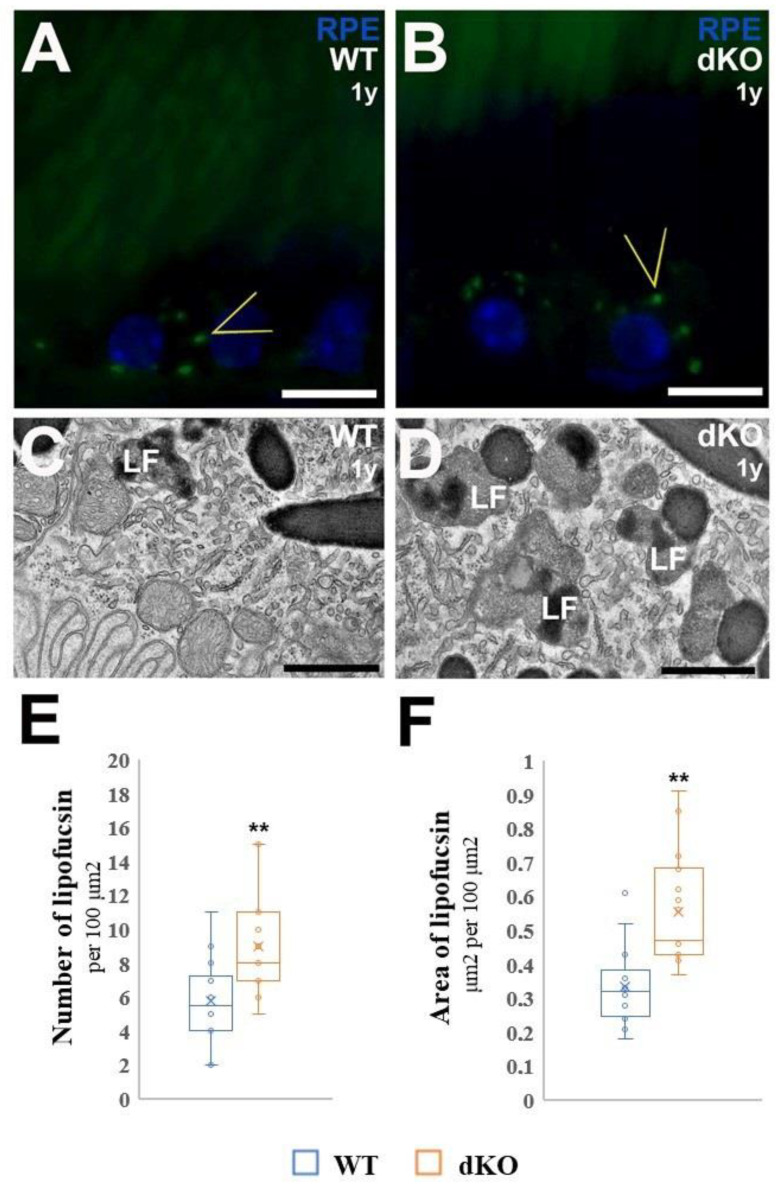
Confocal microscopy of lipofuscin (LF) autofluorescence and quantitative TEM analysis of lipofuscin granules in wild-type (WT) and dKO RPE. Sections from WT and dKO mice were analyzed by fluorescence microscopy. Arrowheads point to LF granules within RPE cytoplasm of one-year-old WT (**A**) and dKO mice (**B**). In contrast to WT, dKO RPE showed increased levels of LF fluorophores. Transmission electron microscopy was used for quantitative analysis of polymorphic lipofuscin granules in WT (**C**) and dKO (**D**) samples. Considerable accumulation of the number of LF granules was detected in dKO samples. The lipofuscin granules in the cytoplasm of RPE and decrease of organelles were observed; most LF granules resulted to be completely occupied by strongly electron-dense substance (**E**). The average area of LF granules increased within the RPE of dKO samples per 100 μm^2^ area measured from electron micrographs (**F**). ** *p* < 0.001 (Mann-Whitney U-test). Scale bars: 5 μm (**A**,**B**), 0.5 µm (**C**,**D**).

**Figure 7 ijms-22-01684-f007:**
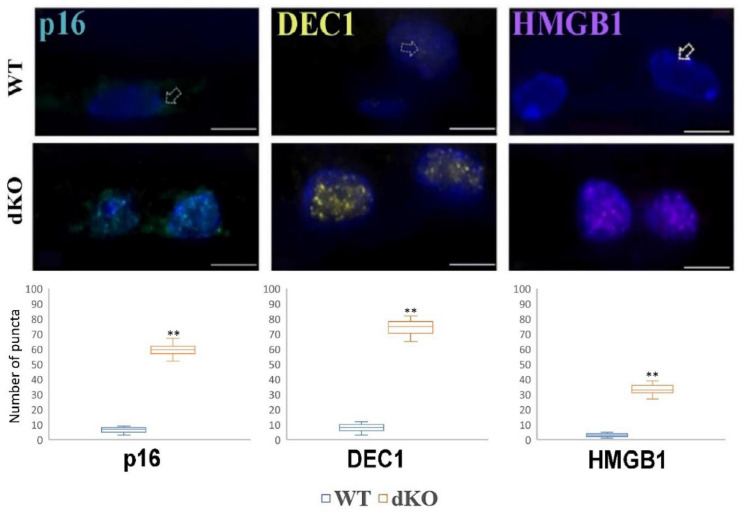
Confocal microscopy analysis of the immunoreactivity of the senescence markers p16, DEC1 and HMGB1 in the retinal pigment epithelium (RPE) of one-year-old *NFE2L2/PGC-1α* dKO mice as compared with their WT counterparts. Semi-quantitative comparative densitometric analysis performed by ImageJ of the immunoreactivity of the markers in 1-μm parasagittal sections of RPE. Alexa Fluor secondary antibodies were used to detect senescence markers. DAPI was used to stain the nuclei of RPE cells. Scale bar: 5 μm. Arrows indicate staining of senescence markers in the RPE. ** *p* < 0.001 (Mann-Whitney U-test).

**Figure 8 ijms-22-01684-f008:**
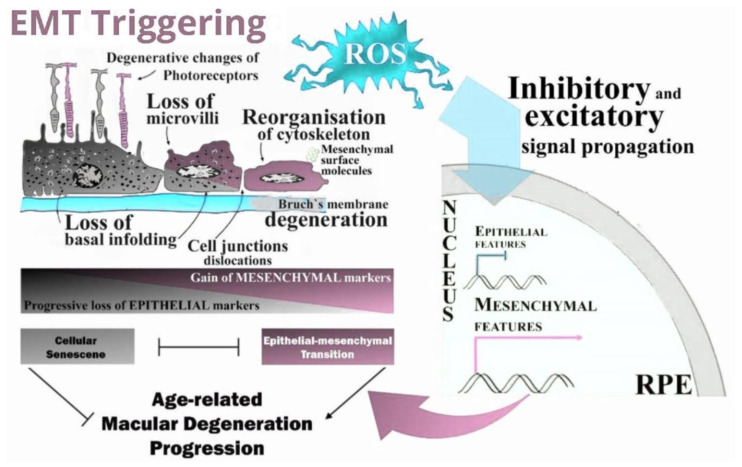
ROS-induced EMT-triggered signal propagation and transduction: an intricate interplay in the RPE of *NFE2L2*/*PGC-1α* double knockout model enhances epithelial-mesenchymal transition (EMT). Simplified representation of the oxidative stress (OS)-dependent major molecular mechanisms of EMT. Reprogramming of gene expression during EMT, and non-transcriptional changes, are initiated and controlled by signaling pathways that respond to extracellular cues. Here, the arrows pointing from the signaling routes to the EMT transcription factors in the nucleus are grouped. EMT involves a functional transition of highly polarized retinal epithelial cells into ECM component-secreting mesenchymal-type flattened cells. Identification of all mesenchymal cells originating from the RPE via EMT is hardly possible as many mesenchymal cells display some epithelial markers while a transition is not fully completed.

**Table 1 ijms-22-01684-t001:** Antibodies used in immunohistochemical analysis.

Antibody Against	Supplier (Catalog Number)	Dilution
Collagen 1	sc-393573	1:100
Vimentin	sc-373717	1:100
OB-Cadherin	sc-365867	1:100
Slug	sc-166476	1:100
Snail	sc-271977	1:100
Ubiquitin	z-0458	1:200
Phalloidin	a-12380	1:100
Dec1	ls-B2222-100	1:100
p16	ab-189034	1:100
HGMB1 (high mobility group box 1)	ab-79823	1:100

## Data Availability

Data available on request from the corresponding author.

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
