# Peer review of "Epithelial-Mesenchymal Transition and Senescence in the Retinal Pigment Epithelium of *NFE2L2/PGC-1α* Double Knock-Out Mice"

_ijms, 2021, doi:10.3390/ijms22041684_

Round 1

Reviewer 1 Report

I have no additional comments at this point.

Reviewer 2 Report

I think the revision improves the quality of manuscript.

This manuscript is a resubmission of an earlier submission. The following is a list of the peer review reports and author responses from that submission.

Round 1

Reviewer 1 Report

It would be better to include the single KO mice as a comparison to DKO mice. More robust quantitative methods for EMT and senescence markers are critical. Several time points of disease-like phenotypes would be significant to support the conclusions.

Author Response

Comment: It would be better to include the single KO mice as a comparison to DKO mice. More robust quantitative methods for EMT and senescence markers are critical. Several time points of disease-like phenotypes would be significant to support the conclusions.

Answer: In this manuscript we presented an extension of AMD-related effects that can be studied with our model of AMD based on dKO, not sKO mice. The comparison of dKO vs. sKO animals was done in our previous Redox Biology paper. Now we have one year old dKO and NEF2L2 sKO animals but our PGC-1α sKO mice are 3 months old. Surely, more methods and extended time frame can be applied in further research to have a complete and more convicing picture of the effects ongoing in the model.

Reviewer 2 Report

This manuscript describes EMT and senescence of RPE in NRF-2/PGC1a double knockout mouse. Authors showed that pathologic structural alterations and changes in EMT and senescence markers in immunostaining analysis. These approaches seem to be originated from their previous publication in Redox biology showing that NRF-2/PGC1a double knockout induces oxidative stress, mitochondrial dysfunction, ER stress and suppresses autophagic degradation in RPE. However, this manuscript does not provide further interesting findings and supporting mechanisms related to EMT and senescence in the absence of NRF-2/PGC1a in RPE. Furthermore, only immunostaining observation may not be enough to suggest their hypothesis.

Author Response

Comment: This manuscript describes EMT and senescence of RPE in NRF-2/PGC1a double knockout mouse. Authors showed that pathologic structural alterations and changes in EMT and senescence markers in immunostaining analysis. These approaches seem to be originated from their previous publication in Redox biology showing that NRF-2/PGC1a double knockout induces oxidative stress, mitochondrial dysfunction, ER stress and suppresses autophagic degradation in RPE. However, this manuscript does not provide further interesting findings and supporting mechanisms related to EMT and senescence in the absence of NRF-2/PGC1a in RPE. Furthermore, only immunostaining observation may not be enough to suggest their hypothesis.

Answer: The present study is a continuation of the research reported in Redox Biology with the use of the same dKO mouse model of AMD. Neither EMT nor senescence were explored in our previous research, nor has it been researched in other laboratories with such a model. Our immunostaining observations are supported by transmission electron microscopy results.

Reviewer 3 Report

Age-related macular degeneration (AMD) is a chronic and progressive eye disease leading to blindness, that currently affects 50 million Europeans over the age of 30. Relevant clinical models of disease are lucking, and a better understanding of the pathophysiology of the disease is required for more effective and long-lasting treatment outcomes. The work by Blasiak et al., shows that knock out of NFE2L2/PGC-1α gene leads to morphological changes in the retina, which changes mimic the pathophysiology of the dry form of AMD, and that these changes translate into breakdown of the blood-retinal barrier, and with enhanced Epithelial-mesenchymal transition. 

Comments to authors

  1. The article should be revised for grammatical errors for example the following statement in the figure legend: ‘’In comparison, in NFE2L2/PGC-1α dKO samples RPE height was bit less variability, due to do the higher presence of more atrophic ones at this special localization in dKO samples (E)’’.

  1. A paragraph should be included in the introduction highlighting the current challenges facing treatment of AMD, and why additional research is needed, and for what?

  1. Figure 1 A and B, please use images of the same magnification in both the WT and in the dKO. Inserts zooming into areas of interest can then be added separately to emphasize the details intended. In the figure legend please explicitly state what the error bars represent. What does the insert in figure 1A represent? What do the black and red arrow heads represent in panels C and D? There is enough space in figure 1, please revise the figure to enlarge panel E making the image more visible. Axis markings should be included, and error bars made more visible. In E, if the RPE thickness was indeed normalized to the WT, then how was the fold change determined? please explain this. What statistical test was performed to determine if the change was significant or not? This should be stated in the figure legend.

  1. The following statement is inaccurate ‘’The magnified high-power images show irregular intercellular apical junctions in dKO RPE cells (Figure B) compared to typical hexagonal RPE morphology in WT mice (Figure 1A)’’ given that first the images are incomparable due to the different magnification size, and in addition, the image in 1A has a bit of irregular structures which are even evident at low magnification. Please provide more convincing evidence to support this claim.

  1. Figure 2. Whenever comparing WT to dKO, please use images with the same magnification in both the WT and in the dKO. Intensity fold change is not accurate enough to support the claim an increased immunoreactivity of the mesenchymal markers Snail, Slug, vimentin, and OB-cadherin in the retina of dKO. This data should be backed-up by additional evidence from more quantitative methods such as Western blot analysis, and or by qPCR for selected factors. For each confocal micrograph in figure 2, please provide lettering for ease of interpretation of the figure. Also, authors consider using an alternative color for OB-CAD instead of white, as the white color is already assigned to label ‘’WT and dKO’’. Authors state that the scale bar – 5 µm, however, there is no scale bar for the image for SLUG, VIM and Cil1. Please verify this. Just like in figure 1, What statistical test was performed to determine if the change was significant or not? This should be stated in the figure legend.

  1. Please elaborate in detail how the transgenic mice were generated. Was the KO conditional? Was it localized to a given cell type? In addition, please provide the genotyping data of the animals at least as a supplementary material. Please justify the use of the Mann-Whitney U-test for statistical analysis.

Author Response

Comment: The article should be revised for grammatical errors for example the following statement in the figure legend: ‘’In comparison, in NFE2L2/PGC-1α dKO samples RPE height was bit less variability, due to do the higher presence of more atrophic ones at this special localization in dKO samples (E)’’.

Answer: We have done our best to correct all linguistic errors.

Comment: A paragraph should be included in the introduction highlighting the current challenges facing treatment of AMD, and why additional research is needed, and for what?

Answer: We have added the following fragment to the Introduction section:

“Current AMD therapy faces many challenges. The most prevalent form of AMD, dry AMD (85% of cases), is currently untreatable.  Only wet AMD can be treated with intravitreal anti-vascular epithelial growth factor (VEGF) inhibitors that slow disease progression. The life-long monthly injections are a burden to the patients and healthcare systems.  Moreover, intravitreal injection is an invasive procedure with some potential side effects, such as hemorrhages and inflammatory reactions. Total worldwide costs for AMD treatment are estimated to be 350 billion USD per year [22].

Some successes of gene therapy and stem cell-based treatment have not been implemented in the clinical practice. Retinal drug therapy is hampered by ineffective and/or short-acting drug delivery to the targets. Eye drop instillation does not provide sufficient drug bioavailability and the blood-retina barrier limits the penetration of systemic drugs. For most drugs, intravitreal injections are not useful because small molecules are eliminated from the eye within 2-3 days. Protein-based drugs also have a narrow therapeutic index, and they cannot reach their biological targets [23]. Therefore, the treatment of AMD is very limited, and, in many cases, it can be considered as an incurable disease. One of the reasons for this is the complexity of AMD pathogenesis and our incomplete knowledge of its underlying mechanisms. Therefore, studies on molecular aspects of AMD pathogenesis are needed as their results can contribute to the identification of new therapeutic targets.”

with new references:

  1. AMD Alliance International. http://www.brightfocus.org/sources-macular-degeneration-facts-figures (accessed December 18, 2020).
  2. Del Amo, E.; Vellonen, K-S,; Kidron, H.; Urtti, A. Intravitreal clearance and volume of distribution of compounds in rabbits: In silico prediction and pharmacokinetic simulations for drug development. European Journal of Pharmaceutics and Biopharmaceutics 2015, 95,215-26. doi: 10.1016/j.ejpb.2015.01.003.

Comment: Figure 1 A and B, please use images of the same magnification in both the WT and in the dKO. Inserts zooming into areas of interest can then be added separately to emphasize the details intended.

Answer: Figure 1 A and B are now at the same magnification including scale bars.

Comment: In the figure legend please explicitly state what the error bars represent.

Answer: Error bars have been defined now in Figure legends.

Comment: What does the insert in figure 1A represent?

Answer: We deleted the schematic presentations of the localization of tight junctions in the RPE shown earlier in the insert figure 1A.

Comment: What do the black and red arrow heads represent in panels C and D? There is enough space in figure 1, please revise the figure to enlarge panel E making the image more visible. Axis markings should be included, and error bars made more visible. The following statement is inaccurate ‘’The magnified high-power images show irregular intercellular apical junctions in dKO RPE cells (Figure B) compared to typical hexagonal RPE morphology in WT mice (Figure 1A)’’ given that first the images are incomparable due to the different magnification size, and in addition, the image in 1A has a bit of irregular structures which are even evident at low magnification. Please provide more convincing evidence to support this claim.

Answer: We have changed Figure 1 and its legend according to these comments. Please see our new Figure 1 and its legend.

Comment: In E, if the RPE thickness was indeed normalized to the WT, then how was the fold change determined? please explain this. What statistical test was performed to determine if the change was significant or not? This should be stated in the figure legend.

Answer: We removed “Fold change” from the title of the ordinate axis in Figure 1 and changed the figure legend accordingly. We added information on the test we applied to determine significance of the differences between WT and dKO animals.

Comment: Figure 2. Whenever comparing WT to dKO, please use images with the same magnification in both the WT and in the dKO. Intensity fold change is not accurate enough to support the claim an increased immunoreactivity of the mesenchymal markers Snail, Slug, vimentin, and OB-cadherin in the retina of dKO. This data should be backed-up by additional evidence from more quantitative methods such as Western blot analysis, and or by qPCR for selected factors. For each confocal micrograph in figure 2, please provide lettering for ease of interpretation of the figure. Also, authors consider using an alternative color for OB-CAD instead of white, as the white color is already assigned to label ‘’WT and dKO’’. Authors state that the scale bar – 5 µm, however, there is no scale bar for the image for SLUG, VIM and Cil1. Please verify this. Just like in figure 1, What statistical test was performed to determine if the change was significant or not? This should be stated in the figure legend.

Answer: We have modified Figure 2 and its legend according to the above comments, but we are not able to perform either qPCR or WB analysis within a reasonable time due to animal breeding, cell isolation and culturing, which are time-consuming procedures.

Comment: Please elaborate in detail how the transgenic mice were generated. Was the KO conditional? Was it localized to a given cell type? In addition, please provide the genotyping data of the animals at least as a supplementary material.

Answer: We applied unconditioned, global NFE2L2/PGC-1α knockout and all details of the generation of the transgenic mice were described in our previous publication in Redox Biology as the animals were from the same stock.

Comment: Please justify the use of the Mann-Whitney U-test for statistical analysis.

Answer: First, sorry for the error we made in presenting our data – neither mean, nor SD should have been presented. The use of the Mann-Whitney U-test was justified by (1) low number of samples and (2) distribution departing from normality. We have changed the Data analysis section into:

“Normality of data distribution was checked using Shapiro–Wilk W test. As the distribution of data departed from the normal distribution, the Mann-Whitney U test was used. Data were presented as median with I and III quartiles and error bars representing 1.5 times the interquartile distance. Statistical analyses were performed with IBM SPSS Statistics for Windows (version 24.0, Armonk, NY, USA).”

Round 2

Reviewer 1 Report

The manuscript relies solely on the immunofluorescence-based quantification method. Although such a semi-quantitative way is informative but not conclusive, some other supportive approaches are essential to support the study’s hypothesis. The data included in the manuscript are inadequate to draw the conclusions that the authors claimed.

Reviewer 2 Report

I understand the explanation by authors, and I agree to accept their revised version for publication.

(* It was difficult for me to find the number of mouse (N) used for each figure in statistical analysis.)